

# UDP-glucose pyrophosphorylase: genome-wide identification, expression and functional analyses in *Gossypium hirsutum*

Zhongyang Xu[1], Jiasen He[1], Muhammad Tehseen Azhar[2], Zhen Zhang[3], Senmiao Fan[3], Xiao Jiang[3], Tingting Jia[3], Haihong Shang[1] and Youlu Yuan[3]

[1] Zhengzhou Research Base, State Key Laboratory of Cotton Biology, School of Agricultural Sciences, Zhengzhou University, Zhengzhou, Henan, China

[2] Department of Plant Breeding and Genetics, University of Agriculture, Faisalabad, Pakistan

[3] State Key Laboratory of Cotton Biology, Key Laboratory of Biological and Genetic Breeding of Cotton, The Ministry of Agriculture, Institute of Cotton Research, Chinese Academy of Agricultural Sciences, Anyang, Henan, China

## ABSTRACT

In this study, a total of 66 UDP-glucose pyrophosphorylase (UGP) (EC 2.7.7.9) genes were identified from the genomes of four cotton species, which are the members of Pfam glycosyltransferase family (PF01702) and catalyze the reaction between glucose-1-phosphate and UTP to produce UDPG. The analysis of evolutionary relationship, gene structure, and expression provides the basis for studies on function of *UGP* genes in cotton. The evolutionary tree and gene structure analysis revealed that the *UGP* gene family is evolutionarily conserved. Collinearity and Ka/Ks analysis indicated that amplification of *UGP* genes is due to repetitive crosstalk generating between new family genes, while being under strong selection pressure. The analysis of *cis*-acting elements exhibited that *UGP* genes play important role in cotton growth, development, abiotic and hormonal stresses. Six *UGP* genes that were highly expressed in cotton fiber at 15 DPA were screened by transcriptome data and qRT-PCR analysis. The addition of low concentrations of IAA and GA3 to ovule cultures revealed that energy efficiency promoted the development of ovules and fiber clusters, and qRT-PCR showed that expression of these six *UGP* genes was differentially increased. These results suggest that the *UGP* gene may play an important role in fiber development, and provides the opportunity to plant researchers to explore the mechanisms involve in fiber development in cotton.

# INTRODUCTION

UDP-glucose pyrophosphorylase (UGP) is a member of Pfam glycosyltransferase family (PF01702), an enzyme found in various organisms, including plants, animals and bacteria (*Chen et al., 2007*; *Johansson et al., 2002*; *Kleczkowski, 1994*; *Winter & Huber, 2000*), which catalyzes the reaction between glucose-1-phosphate and UTP to produce UDPG (*Daran et al., 1995*). UDPG is an important molecule in biology, food, biopharmaceutical and

Corresponding authors
Haihong Shang,
shanghaihong@caas.cn
Youlu Yuan, yuanyoulu@caas.cn,
yylcri@126.com

cosmetic chemistry, and an essential glucose donor compound. It is also one of the key precursors for sugar interconversion, disaccharide and polysaccharide formation, and amino and nucleotide sugar metabolism (*Lamerz et al., 2006*), and is involved in several essential cellular processes, including carbohydrate metabolism, cell wall biosynthesis, and protein glycosylation (*Chen et al., 2007*; *Daran et al., 1995*).

Previous studies have shown that *UGP* genes have diversity of roles in various organisms. For example, in fungi, UDPG is an essential precursor of b-1,3-glucan and b-1,6-glucan, where both are the components of biosynthesis of cell wall (*Daran et al., 1995*). In yeast, YKL248 cells, UDPG concentration was reduced by 50% when UGP activity was significantly reduced by 10-fold, resulting in the induction of multiple outgrowth phenotypes (*Daran, Bell & François, 1997*; *Daran et al., 1995*). It was also reported that antisense repression of *UGP* genes in plants reduces the content of soluble carbohydrates, starch or sucrose (*Borovkov et al., 1996*; *Spychalla et al., 1994*). In *Arabidopsis*, the *AtUGP1/AtUGP2* double-silent mutant showed the decreased concentration of UDPG, growth defects, and male-sterility (*Park et al., 2010*). Changes in the cell wall structure and number of mycelial meristems in UGP homolog knockout mutants of *Ganoderma lucidum* (*Li et al., 2015b*). In rice, silencing of *UGP* 1 by co-repression or double-stranded RNA interference (dsRNAi) affects the callus deposition during meiosis of pollen, resulting in male sterile phenotype (*Chen et al., 2007*; *Woo et al., 2008*). The data also suggests that over-expression of native or exotic *UGP* gene in various plants can the increase plant height, leaf area and leaf-stem biomass ratio or nutritional profile (*Coleman et al., 2006*; *Payyavula et al., 2014*).

Cotton is one of the most important sources of fiber in the world. The widely cultivated upland cotton (*Gossypium hirsutum*) is an allotetraploid originated from two diploid ancestral species, *G. arboreum* (A-genome) and *G. raimondii* (D-genome), resulting from natural hybridization and genome doubling over millions of years in natural conditions (*Wendel, 1989*; *Wendel & Cronn, 2003*). Amongst the several quality traits, fiber strength is one of the important traits, where cotton fiber is a single-celled seed hair of the ovule epidermis, whose development is accomplished in four different stages (*Kim & Triplett, 2004*), in which cell elongation determines the primary quality traits of cotton fiber (*Deng et al., 2012*). Fibroblast elongation is a complex process involving multiple metabolic and regulatory events (*Kim & Triplett, 2001*). It was reported in literature that various abiotic stresses and hormonal homeostasis play a crucial role in the development and quality of cotton fibers, such as BR (*Sun et al., 2005*; *Yang et al., 2014*). Fiber strength is mainly determined by the strength of its cell wall, where main components of cell wall includes cellulose and non-cellulose components. Cellulose is formed by UDPG, and UDPG can be synthesized by employing three enzymes namely, UGP, UDP-sugar-pyrophosphorylase (USP*)* and sucrose synthase. For the synthesis of UDPG and UGP, USP use monosaccharide-1-phosphate as a substrate, while SuSy catalyzes sucrose cleavage and delivers UDPG directly to the plasma membrane-associated cellulose synthase complex (*Amor et al., 1995*; *Kotake et al., 2004*). These findings support the existence and important role of *UGP* genes in fiber development of cotton, and the sequencing of cotton genome

has made it possible to analyze various gene families through genome-wide approach (*Du et al., 2018*; *Hu et al., 2019*; *Paterson et al., 2012*; *Zhang et al., 2015*).

In this study, 19 *UGP* genes were identified form upland cotton through gene family identification, phylogenetic tree construction, structural analysis, chromosome distribution, analysis of covariance and its Ka/Ks ratio, prediction and analysis of promoter *cis*-acting elements, transcriptome data analysis, ovule culture and its phenotypic observation, *etc*. The expression pattern of *UGP* gene in cotton fiber was analyzed by using qRT-PCR. In addition, we have determined the expression patterns of *UGP* genes under phytohormone-stimulated conditions to explore the functional role of these genes in cotton. This study provides the foundation for elucidation of evolutionary and functional analysis of *UGP* genes and provides a molecular and biological basis for a deeper understanding of the association between *UGP* genes and cotton fiber development.

## MATERIALS AND METHODS

### Identification of *UGP* gene family

The *UGP* gene was obtained by searching the whole genome (TAIR: http://www.arabidopsis.org) of Arabidopsis thaliana, and Hidden Markov Model (HMM) of *UGP* (PFAM01704) was obtained by searching the conserved structural domain of proteins through National Center for Biotechnology Information Search database (NCBI), and the HMM was used in the hmmer search program in hmm3.0 software (*Finn, Clements & Eddy, 2011*). The genomic data of seven species were screened out using the Arabidopsis UDPGP model and downloaded from the CottonFGD (https://cottonfgd.org/about/download.html) website for *G. arboreum* (Ga) (*Du et al., 2018*), *G. barbadense* (Gb) (*Hu et al., 2019*), *G. hirsutum* (Gh) (*Hu et al., 2019*) and *G. raimondii* (Gr) (*Paterson et al., 2012*), *Theobroma cacao* (*Motamayor et al., 2013*) and *Carica papaya* (*Ming et al., 2008*) were retrieved from ePhytozome 2.1 database6 databases (https://phytozome-next.jgi.doe.gov/) the *Arabidopsis thaliana* genome sequence (*A. thaliana*) was retrieved from the TAIR database. The search results were filtered with a threshold value of $E$-value $= 1e-5$, and unqualified and duplicate transcripts were discarded after the results were obtained.

### Construction and structural analysis of the phylogenetic tree of *UGP* gene

The protein sequences of seven species were compared by using MEGA7.0 "muscle" analysis (*Kumar, Stecher & Tamura, 2016*), and then neighbor-joining method (NJ) (*Saitou & Nei, 1987*) was used to generate a phylogenetic tree with a bootstrap of 1000 (*Tamura et al., 2013*). In addition, the sequences of *UGP* based from four species of cotton were isolated to construct an evolutionary tree *UGP* gene, and intron-exon analysis was performed by using TBtools (*Chen et al., 2020*). The gene motifs were analysed through the MEME website (meme-suite.org/meme/index.html), and a minimum of 10 motifs were designed (*Hu et al., 2015*). The isoelectric point (PI) and molecular weight (MW) of these genes were calculated separately online through the ExPASy website (http://web.expasy.org/compute pi/).

### Chromosome location, collinearity and Ka/Ks analysis

TBtools (*Chen et al., 2020*) were applied on cotton genome file and GFF3 file to draw the position of gene on the chromosome. Homologous *UGP* gene pairs were obtained by BLASTP full-pair search (*Altschul et al., 1990*), and visualization was obtained by TBtools (*Chen et al., 2020*).

### Prediction and analysis of promoter cis-acting elements

The 2,000 bp sequence form upstream of start codon of the *UGP* gene was extracted from the CottonFGD website, and the cis-regulatory elements were predicted through PlantCARE database (http://bioinformatics.psb.ugent.be/webtools/plantcare/html/) (*Lescot et al., 2002*). The predicted cis-elements were categorized according to the role in transcriptional regulation (*Pandey et al., 2016*).

### Transcriptome data analysis

The transcriptome data of standard genetic line TM-1 was retrieved from NCBI SPA (https://www.ncbi.nlm.nih.gov/bioproject/PRJNA490626) (*Hu et al., 2019*), and transcriptome data of sGK9708 and 0–153 was obtained from Zhang's research group (*Zhang et al., 2020*). The date for *UGP* gene were extracted from three cotton transcriptome databases mentioned above at various time periods for preliminary analysis.

Types of cotton 0–153 and sGK9708 from based at Chinese Academy of Agricultural Sciences, Zhengzhou, China (*Zhang et al., 2020*). The day to flowering was marked as 0 day post anthesis (0 DPA), and cotton bolls with five, 10, 15, 20, 25 and 30 DPA were taken as materials, and stores in liquid nitrogen for various assays. The RNA extraction calibration of RNA concentration, and reverse synthesis of cDNA was prepared according to the protocols provides by manufacturers, and similar protocols were used previously by *Jia et al. (2020)* and final results were analysed and plotted by using available TBtools (*Chen et al., 2020*). In addition, fluorescent quantitative specific primers were designed on Primer3 (bioinfo.ut.ee/primer3-0.4.0/primer3/) and upland cotton GhHistone3 (AF024716) was used as an internal reference gene (*Xu et al., 2004*) in current studies.

### Ovule culture and its phenotypic observation

The response against four growth hormone, namely, indole acidic acid (IAA), gibberellin (GA3), abscisic acid (ABA) and salicylic acid (SA) was determined by 2.4 analysis, , and one additional hormone ethylene (ETH) was added for ovule *ex vivo* culture experiments.

Ovules from cultivars 0–153 were cultured and five hormones, IAA, GA3, ETH, SA and ABA, were added at final concentrations of 0.1 uM, 0.5 uM and 1 uM. Hormone configuration and packaging method; sampling of bolls and ovules, cleaning and sterilization, culture conditions, and fiber cluster area was measured according to methods previously used by *Jia et al. (2020)*. All of experiments were performed in three independent replicates, where, four to five ovules were selected for fiber quality area assessment and were analysed for statistical significance by using a $t$-test.

Intact ovules cultured for 15 DPA were used as material for RNA extraction and data were summarised by reverse transcription followed by qRT-PCR experiments (methods as above). Data were analysed to observe gene expression.

## RESULT

### Identification of the *UGP* gene family

A total of 81 *UGP* genes were identified in seven species by screening genes containing complete sequences for UGP structural domains, including nine from *G. arboreum*, 26 from *G. barbadense*, 19 from *G. hirsutum*, 12 from *G. raimondii*, seven in *A. thaliana*, four in *C. papaya*, and *T. cacao* four. Among them, a total of 66 *UGP* genes were identified from four cotton species (Table S1).

### Evolutional analysis of *UGP* gene family

The *UGP* gene family was divided into two subclades UGP-I and UGP-II based on the evolutionary tree topology, containing 35 and 46 genes, respectively. *Wang et al. (2011)* divided the *UGP* genes into five sub-clades by using an evolutionary tree of *UGP* genes among 11 species, which were differed from previous studies.

Further analysis of conserved structural domains allowed the division of UGP-I into three subgroups UGP-I-A (Fig. 1A), UGP-I-B (Fig. 1B), and UGP-I-C (Fig. 1B), and UGP-II was further divided into three subgroups UGP-II-D (Fig. 1D), UGP-II-E (Fig. 1E), and UGP-II-F (Fig. 1F). In four cotton species, 66 *UGP* genes were divided into these six subgroups, of which A contained 15 genes, B contained seven genes, C contained eight genes, D contained 12 genes, E contained six genes, and F contained 18 genes; while none of gene from *G. arboreum* was found in sub-group C. The number of *G. barbadense* and *G. hirsutum* in each sub-population was about twice as compared to other species. The isoelectric points and molecular weights of these 81 genes were counted, and it was found that ~74% of their isoelectric points were between 5.5 and 7.3, and ~72% of their molecular weights were between 48.6 and 80.6 kD (Table S2).

### Structural analysis of the *UGP* gene family

The *UGP* structural domains were present in all of genes, and *UGPs* were divided into six sub-groups based on length and composition of structural domains, and the conserved structural domains were similar in each sub-group (Fig. 2A). The gene sequence analysis revealed that each sub-group contained similar types and numbers of motif elements, with sub-population F which contain 9–10 motifs, Sub-population C containing only 2–3 motifs which is minimum as compared to others, while remaining sub-populations contained between four and eight motifs, and majority of genes contained motif1 and motif4 (Fig. 2B). Exon analysis revealed that each *UGP* gene contained a high number of exons *i.e.*, 12–20, and only sub-group C had 5–12 exons, and exons were similar with reference to position and length, and UTR structure was found only in *G. raimondii* (Fig. 3).

### Chromosome location and collinearity analysis

The *UGP* genes were evenly distributed on At and Dt sub-genomic chromosomes of upland cotton, where 10 of 19 *UGP* genes were assigned to eight At sub-genomic and nine to seven Dt sub-genomic chromosomes. Two *UGP* genes were present on chromosomes A08, D08, A11 and their D11, respectively, while only one *UGP* gene was contained on A02, A03,

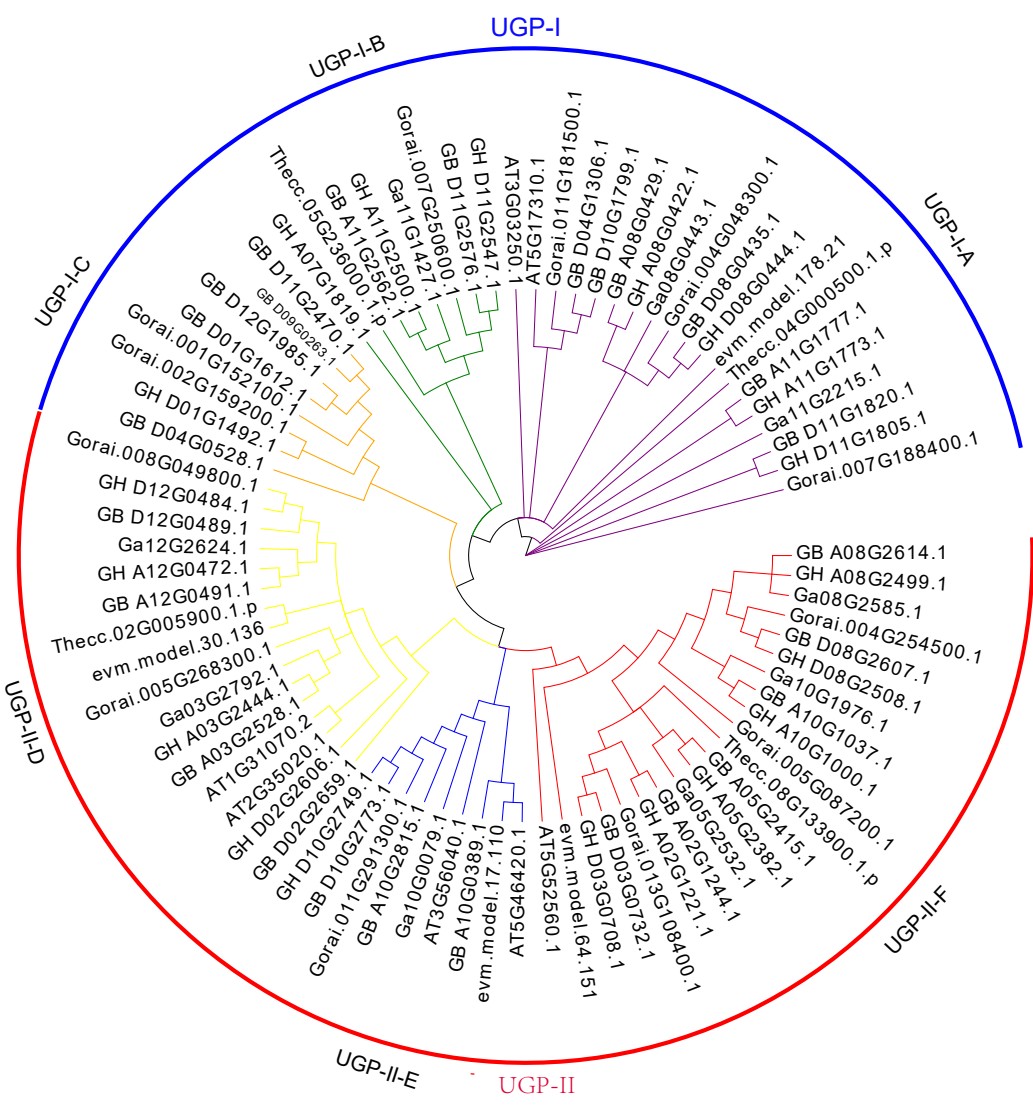

**Figure 1** *UGP* gene phylogenetic tree: A total of 81 *UGP* genes of seven species, including *G. arboretum*, *G. barbadense*, *G. hirsutum*, *G. raimondii*, *A. thaliana*, *C. papaya*, *T. cacao*.

A05, A07, A10, A12, D01, D02, D03, D10 and D12, and no *UGP* gene was found on the remaining chromosomes (Fig. S1).

Because *G. hirsutum* is a four-ploid cotton species formed by natural crosses of two two-ploid cotton species (*G. arboreum* and *G. raimondii*) (*Wendel & Cronn, 2003*). Co-lineage analysis by using 19 *UGP* genes of upland cotton and *G. arboreum* and *G. raimondii* cotton species showed he presence of 47 homologous pairs of genes. 24 homologous pairs were found in A genome and 23 homologous pairs in D genome of upland cotton (Fig. 4A). Co-lineage analysis revealed 45 homologous gene pairs were part of genome of *G. barbadense* (Fig. 4B). The number of *UGP* genes in upland cotton was nearly double that of the two-ploid cotton species, and the At and Dt genomes contained essentially the

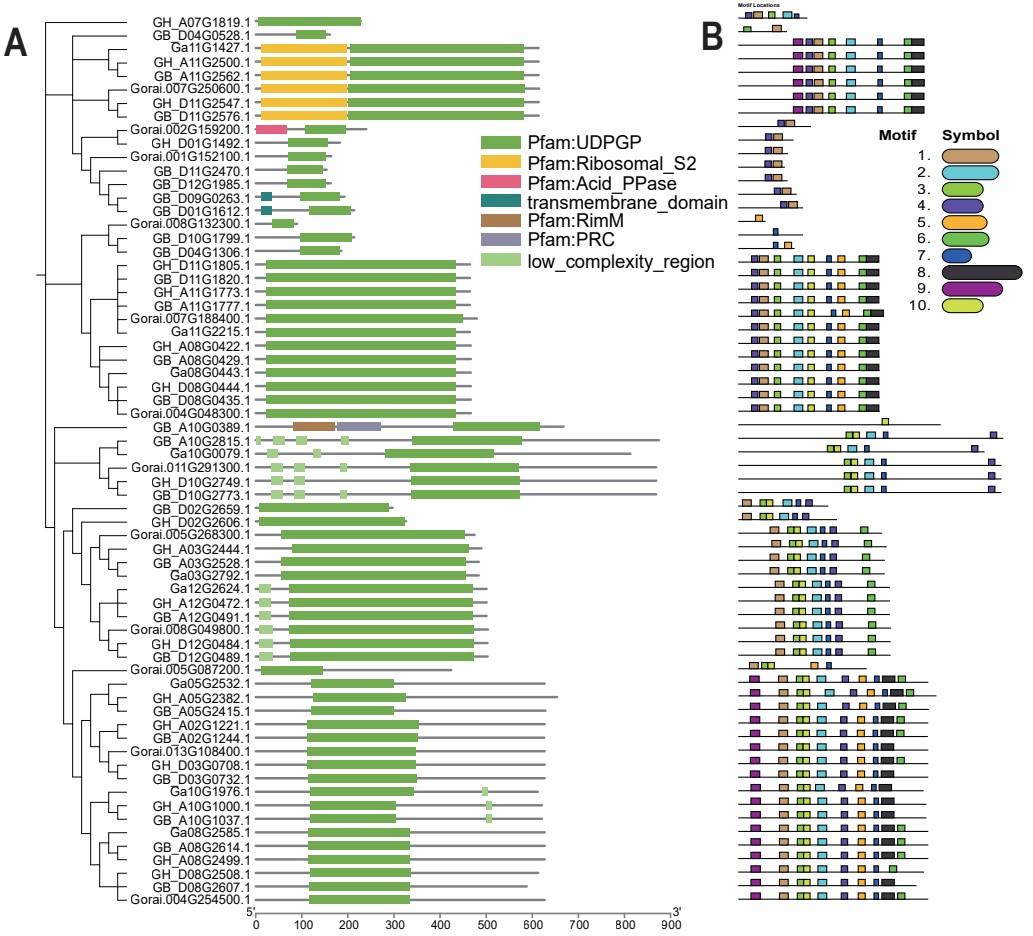

**Figure 2 Evolutionary relationship and motif analysis of cotton *UGD* family.** (A) Phylogenetic tree and conserved domains of *UGP* gene family. (B) Conserved motifs of *UGP* gene family proteins.

same number of homologous gene pairs, suggesting that the *UGP* genes were present in cotton before the upland cotton cross.

The Ka/Ks (non-synonymous/synonymous) ratios homologous gene pairs were calculated by TBtools (*Chen et al., 2020*). It is reported that Ka/Ks = 1.0 represents neutrally selected pseudogenes, Ka/Ks < 1.0 indicates the tendency of purifying selection on replicated genes, and Ka/Ks > 1.0 ratio indicates accelerated evolutionary positive selection (*Qanmber et al., 2019a*). We found that 18 of 19 Gh/Ga homozygous gene pairs, 19 Gh/Gb homozygous gene pairs and 19 Gh/Gr homozygous gene pairs had Ka/Ks values below 1.0, accounting for about 95%; only one gene pair had a Ka/Ks ratio of > 1.0, and since most Ka/Ks were less than 1.0 (Fig. 4C).

## Cotton transcriptome data analysis

Two cotton fiber transcriptome datasets were used to study the expression patterns of the *UGP* gene family: one is NCBI SPA database TM-1 dataset (Fig. 5A) and other is raw RNA-Seq dataset of sGK9708 and 0–153 (Fig. 5B) (Table S3). Analysis of expression

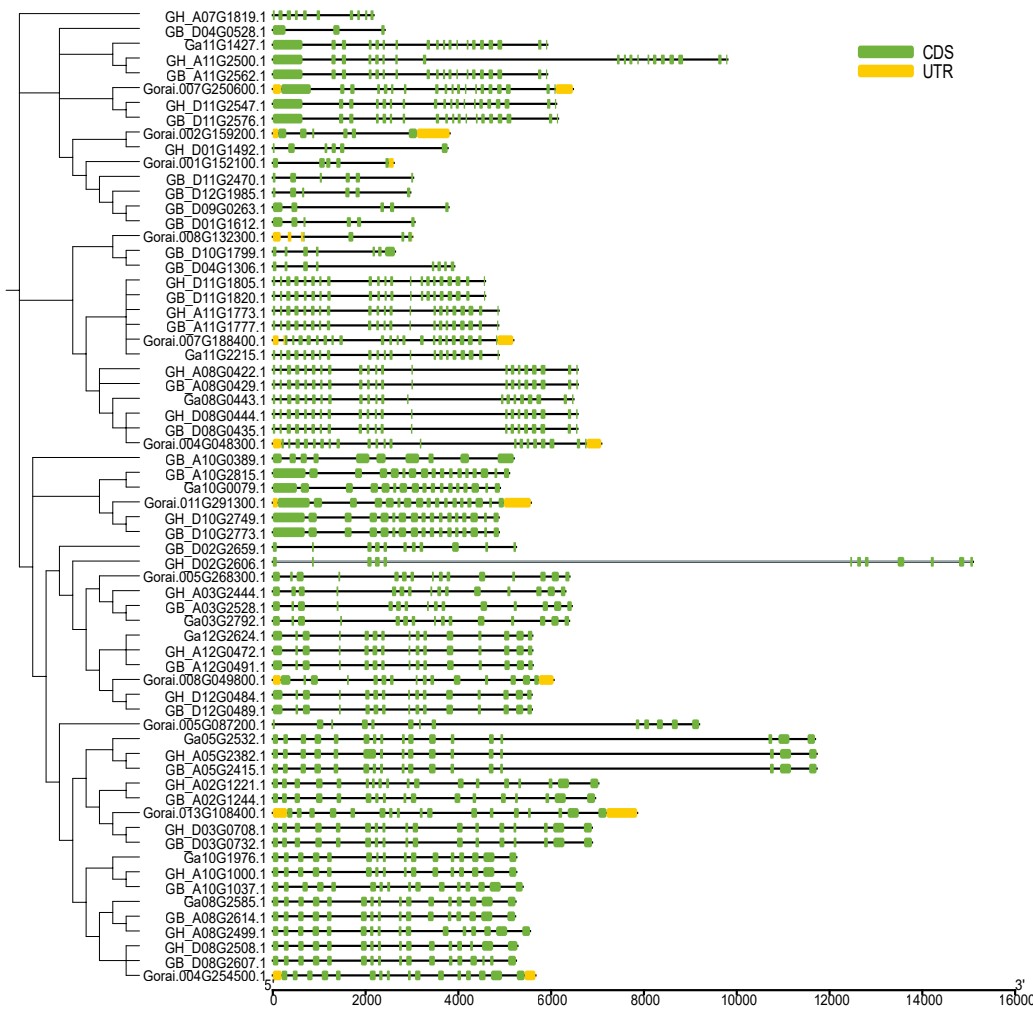

**Figure 3** **Intron distribution of *UGP* gene family.**

data showed that six genes namely *GH_A08G0422*, *GH_D08G0444*, *GH_A12G0472*, *GH_D12G0484*, *GH_A11G1773* and *GH_D11G1805* had relatively high PKFM values in all of three genomes of upland cotton, with highest expression of about 15 days of fiber development. The qRT-PCR data of six genes in days 0–30, showed high expression of 0–153 and sGK9708, which was consistent with the data from transcriptome analysis (Fig. 6).

## Cis-acting regulatory elements in promoter region of *UGP* gene

Analysis of approximately 2,000 bp sequences upstream of start codon (ATG) of 66 genes in *UGP* gene family revealed that all of these genes contain several similar elements, including light and stress response elements, growth response elements (endosperm expression elements, meristem expression elements), and various hormone response elements (Fig. 7). Light response elements include GT1-motif, ACE, G-box, GT1-motif, 3-AF1 binding site and Sp1, four hormone response elements include growth hormone
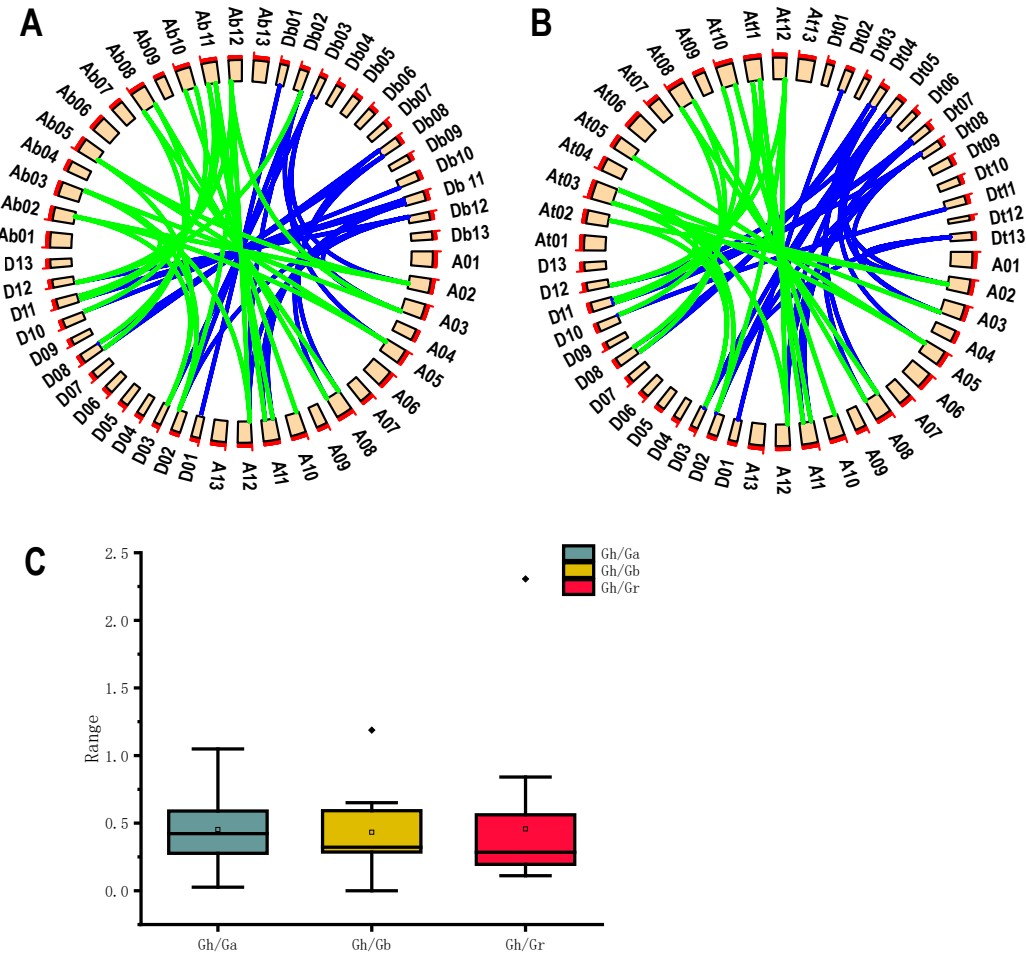

**Figure 4  Collinearity analysis, Ka/Ks analysis and box plot of *UGP* homologous pairs of four cotton species.** (A) Collinearity analysis of GH and GA, GR (B) GH and GB collinearity analysis (C) GH Comparison of Ka/Ks values with the homologous *UGP* family of three other cotton species.

response elements (TGA-element, AuxRR-core), gibberellin response elements (P-box, TATC-box, GARE-motif), salicylic acid response element (TCA-element), and abscisic acid response elements (ABRE), and stress response elements including defense and stress response elements (TC-rich repeats) and low-temperature response elements (LTR). Taken together, the distribution of these different kinds of cis-acting elements responds to the fact that these genes can exert a significant influence on fiber development under light, stress, growth, development, and hormone induction.

## Response of *UGP* genes to different hormones in cotton fibers

Cotton ovules were cultured in three gradients of five hormones, and phenotypic observation and RNA extraction were performed at fiber stage of 15DPA. Six of highly expressed genes (*GH_A08G0422*, *GH_D08G0444*, *GH_A12G0472*, *GH_D12G0484*, *GH_A11G1773*, *GH_D11G1805*) were subjected to qRT-PCR analysis.

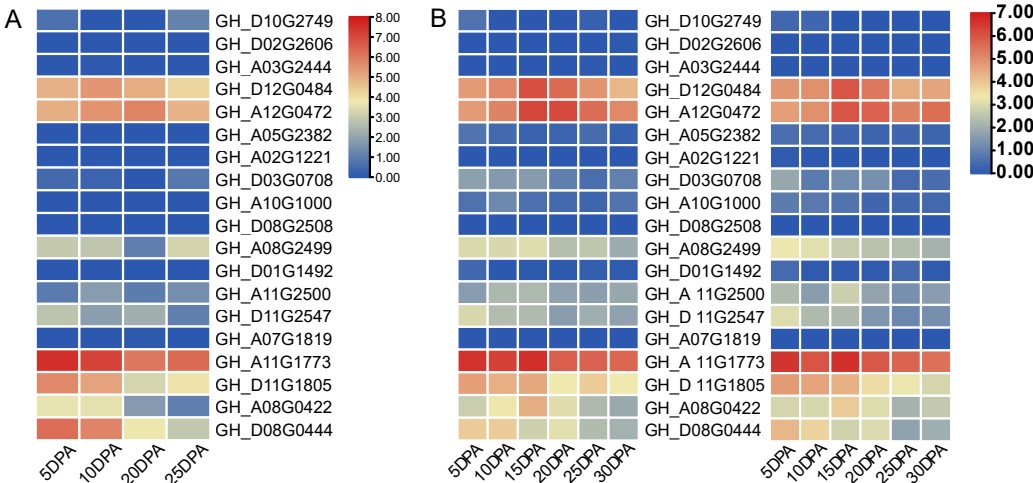

**Figure 5  Heat map of gene expression patterns among different varieties of cotton and gene expression levels in cotton fibers.** (A) Heat map of *UGP* gene expression level in cotton fiber samples collected at 5, 10, 20, 25 DPA in TM-1. Use log2 as the base to scale the expression level. (B) Heat map of *UGP* expression levels in 5, 10, 15, 20, 25, 30 DPA fiber samples of two upland cottons 0–153 and sGK9708.

The supply of IAA and GA3 revealed better growth and development with larger ovule volume and fiber cluster area of 15 DPA ovules than the control, whereas ovules cultures with ABA, ETH and SA grew slowly and had smaller fiber cluster area than the control (Fig. 8A). The area of fiber clusters of ovule culture was counted for each concentration of hormones in a sample of 5–10 ovules, and it was found that fiber cluster development was approximately 10–20% higher than the control with external application of IAA and GA3 hormones, and approximately 20–90% lower under the influence of ETH, SA and ABA hormones (*, $P < 0.05$; **, $P < 0.01$) (Fig. 8B). Analysis of qRT-PCR results of ovule fibers at 15 days showed that the expression of *GH_A08G0422* and *GH_D08G0444* genes were increased differentially with addition of low concentrations of hormones, and were increased to 4–6-fold with addition of IAA. However, the other four genes showed significant increase in expression only when IAA was added (*, $P < 0.05$; **, $P < 0.01$) (Fig. 9).

## DISCUSSION

In the present study, we idefntified 81 *UGP* genes in seven different species, a total of 66 *UGP* genes were identified in four cotton species, which were classified into two subfamilies: UGP-I and UGP-II, containing 46 and 35 genes, respectively, based on topology and conserved structural domains, and the two subfamilies, UGP-I and UGP-II, were divided into three subgroups, respectively. The phylogenetic tree analysis of *UGP* genes from seven different species revealed a very similar homology of *UGP* genes among different species. The distribution of each species in the subpopulations was relatively uniform, demonstrating that *UGP* genes are very conserved across species evolution (*Liu et al., 2018*; *Qanmber et al., 2019a*).
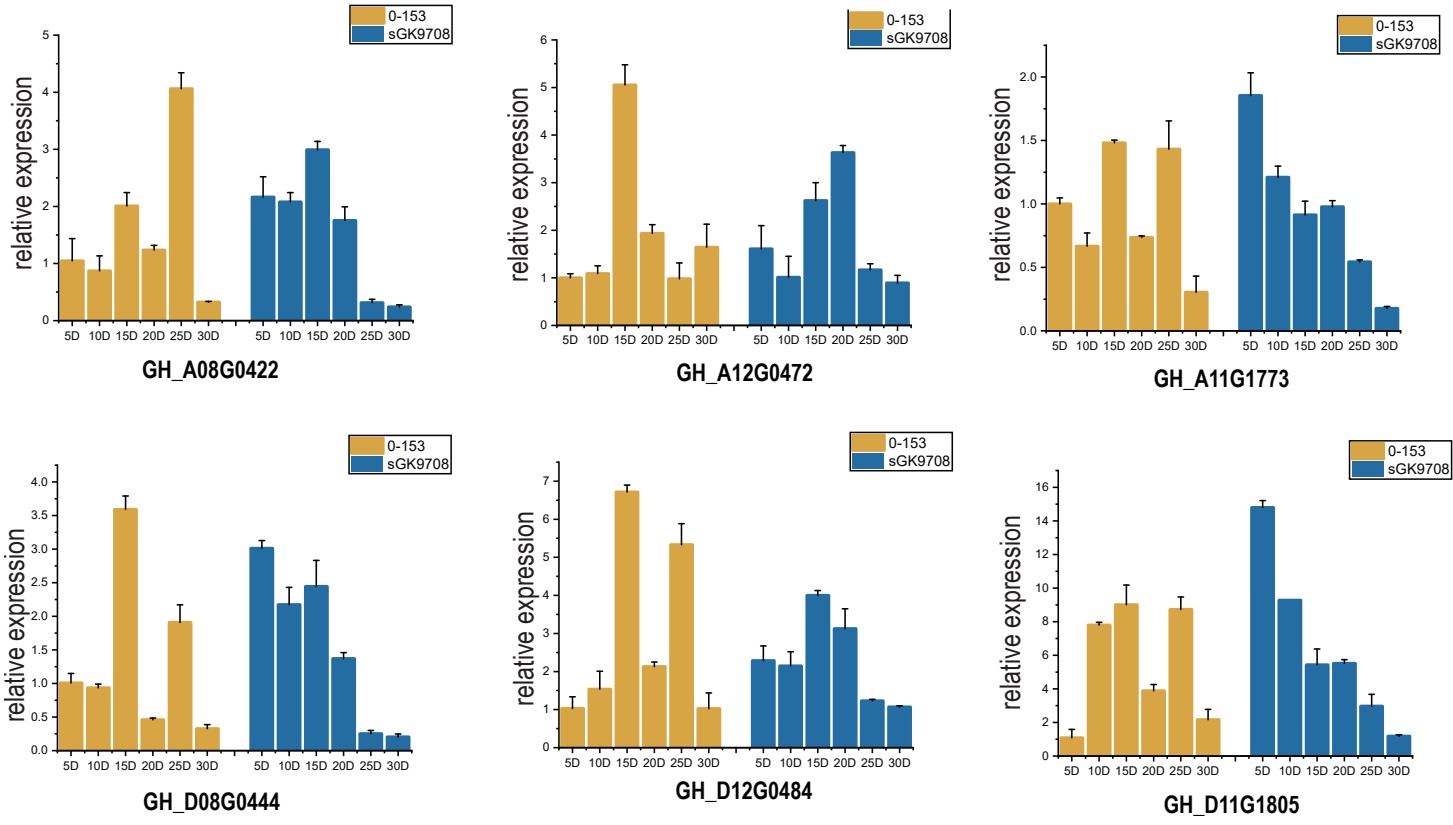

**Figure 6** **Perform qRT-PCR analysis on six *UGP* genes with high expression of 0–153 and sGK9708.** The expression level is shown relative to the internal reference gene GhHis3. Error bars represent the standard deviation of three independent experiments.

The molecular weights of *UGP* genes are concentrated at 48.6–80.6 kD, and most of isoelectric points are concentrated at 5.5–7.3. The same cis-acting elements were found in most of these genes in the analysis of first 2,000 bp cis-acting elements of the start codon, and hormone response elements have been identified, including growth (*Guilfoyle & Hagen, 2007*; *Hagen & Guilfoyle, 2002*) and gibberellin (*Wang et al., 2018*), abscisic acid (*Narusaka et al., 2003*; *Song et al., 2005*)and salicylic acid, and light (*Fankhauser & Chory, 1997*), drought and low-temperature response elements (*Singh, Foley & Oñate Sánchez, 2002*). It is tentatively speculated that *UGP* genes may be involved in growth, development, abiotic and hormonal stresses in cotton (*Qanmber et al., 2019a*; *Qanmber et al., 2019b*).

The *UGP* genes were distributed relatively evenly on the chromosomes, *i.e.,* 10 genes were distributed on At-sub-genic chromosome group and nine genes on Dt-chromosome group. The conserved structural domains, motif analysis and intron analysis revealed that the conserved structural domains were similar in each sub-group with majority of genes containing motif1 and motif4 motif structures, and each *UGP* gene contained a high number of exons between 12–20, and UTR structure was only found in *G. raimondii*. From these finding, it is concluded that *UGP* genes are evolutionarily conserved in cotton (*Qanmber et al., 2018*).

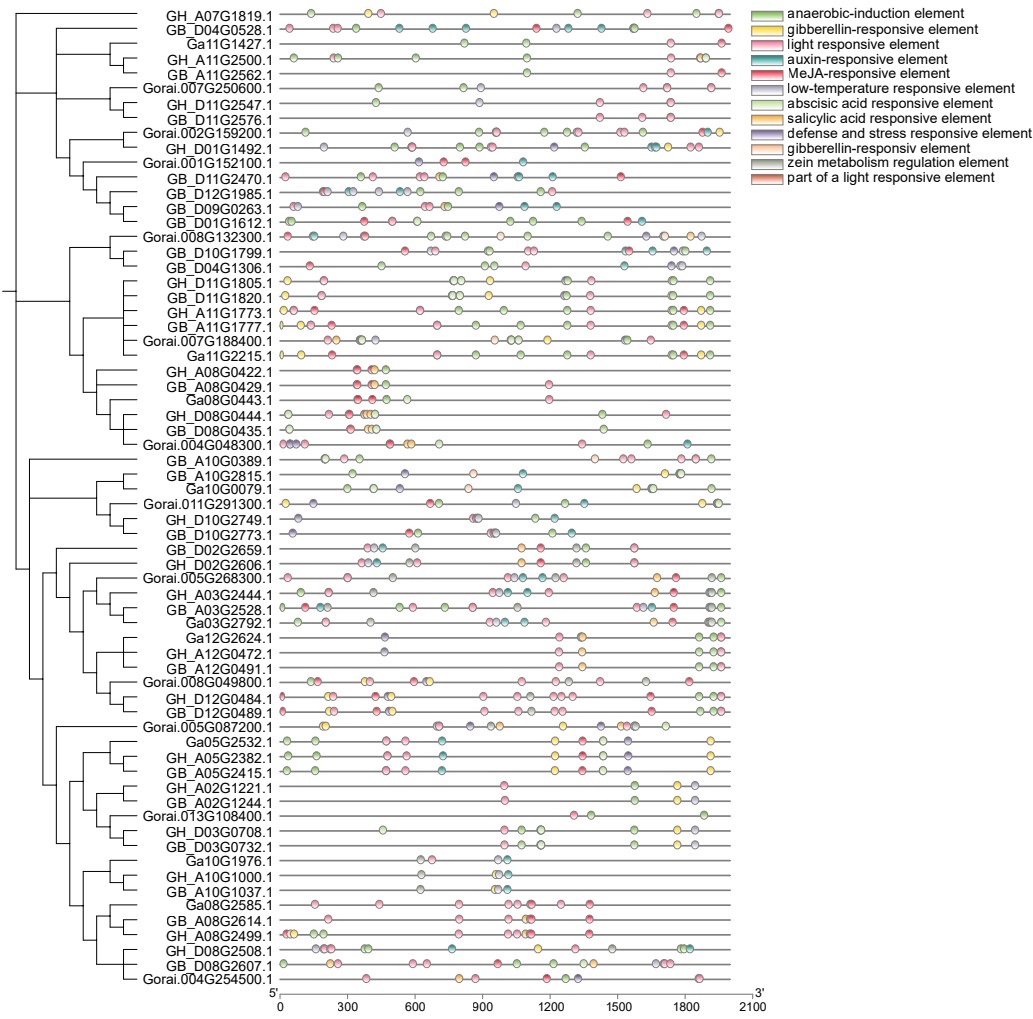

**Figure 7   Analysis of cis-regulatory elements of cotton *UGP* gene.**

The number of *UGP* genes in *G. hirsutum* and *G. barbadense* is about twice as compared to *G. arboreum* and *G. raimondiid*, due to polyploidization. Since *G. hirsutum* was evolved 1.5 million years ago by crossing of diploid ancestral species having A and D genomes (*Hu et al., 2019*; *Li et al., 2015a*; *Schaper & Anisimova, 2015*). It is reported that *UGP* gene was present in parental species of upland cotton. Polyploidy is a common phenomenon in the evolution of plants and is a major mechanism of adaptation and speciation (*Ramsey & Schemske, 1998*). It is estimated that 47–70% of angiosperms are polyploid in nature (*Grant, 1981*; *Masterson, 1994*). Polyploids arises due to involvement of partial or whole-genome duplication (WGD) (*Cannon et al., 2004*). WGD is also a common phenomenon in evolution, and Arabidopsis thaliana has experienced two WGDs that have resulted in DNA loss and chromosomal rearrangements (*Tang et al., 2008*). However, gene loss can occur when genes obtain form duplicate amplification after hybridization (*Li et al., 2015a*; *Paterson, Bowers & Chapman, 2004*), and homologous of *UGP* were lost during evolution

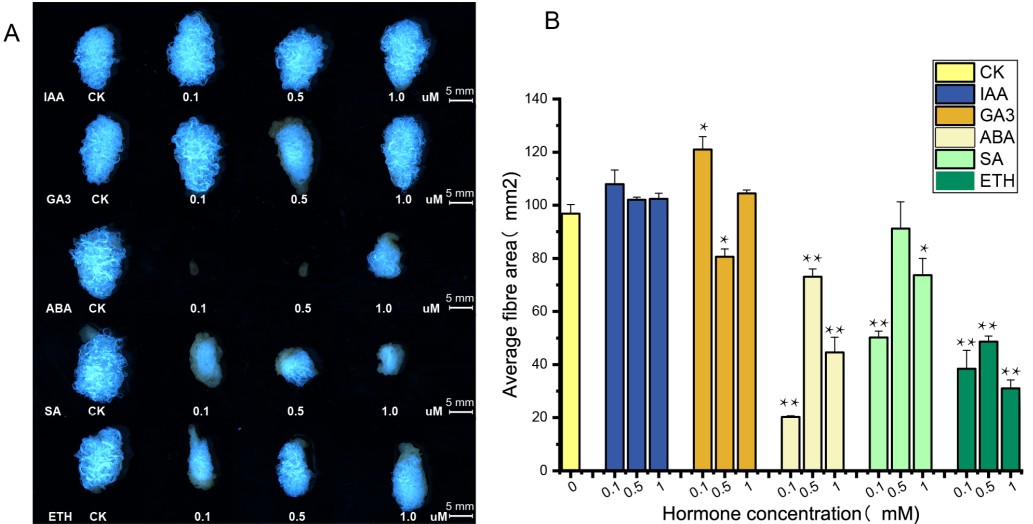

**Figure 8 Phenotypic identification of hormones added to ovule culture.** (A) The phenotypic changes of fiber mass area when hormones of different concentrations are applied in the culture medium. (B) Statistical analysis of phenotypic variation of fiber mass area at different concentrations. *, $P < 0.05$; **, $P < 0.01$.

of upland cotton. This indicated that *UGP* gene is evolutionarily conserved in cotton (*Klinghammer & Tenhaken, 2007*).

The covariance analysis of 19 *UGP* genes found in upland cotton and other three cotton species revealed that most of *UGP* genes were homologous among the four cotton species. Some of homologous genes were prevalent on similar positions, because *UGP* genes were generated from new genes family through tandem repeats during the doubling of upland cotton (*Gě et al., 2020*; *Jia et al., 2020*; *Schaper & Anisimova, 2015*). By calculating the Ka/Ks ratios of homologous gene pairs, it was found that most of Ka/Ks ratios of upland cotton and other three cotton species were below 1.0. This indicated that *UGP* genes were under strong purifying selection pressure during evolution (*Qanmber et al., 2019a*; *Qanmber et al., 2018*).

Nineteen *UGP* genes were analyzed for various timelines of DPA by using transcriptome data, including TM-1, 0–153, and sGK9708. Six genes namely *GH_A08G0422*, *GH_D08G0444*, *GH_A12G0472*, *GH_D12G0484*, *GH_A11G1773* and *GH_D11G1805* were found to be highly expressed in cotton fiber on 15 DPA. RNA from fiber material of cultivar 0–153 and SGK9708 strain was used for qRT-PCR, which supported with transcriptome data (*Zhang et al., 2020*), both *UGP* genes were highly expressed at 15 DPA.

Ovule isolation culture experiments were performed by adding different hormones (*Kim et al., 2015*) and ovules were observed at 15 days of culture. Ovule growth and development were found to be better under due to IAA and GA3, with larger fiber cluster area, while ovule development was retarded under the influence of ABA, ETH and SA. These results indicate that auxin and gibberellin promote cotton ovule growth, while the other three hormones have opposite effects, which are consistent with previous studies (*Wang et al., 2018*; *Zhang et al., 2011*). Ovule samples at 15 days of culture were taken for
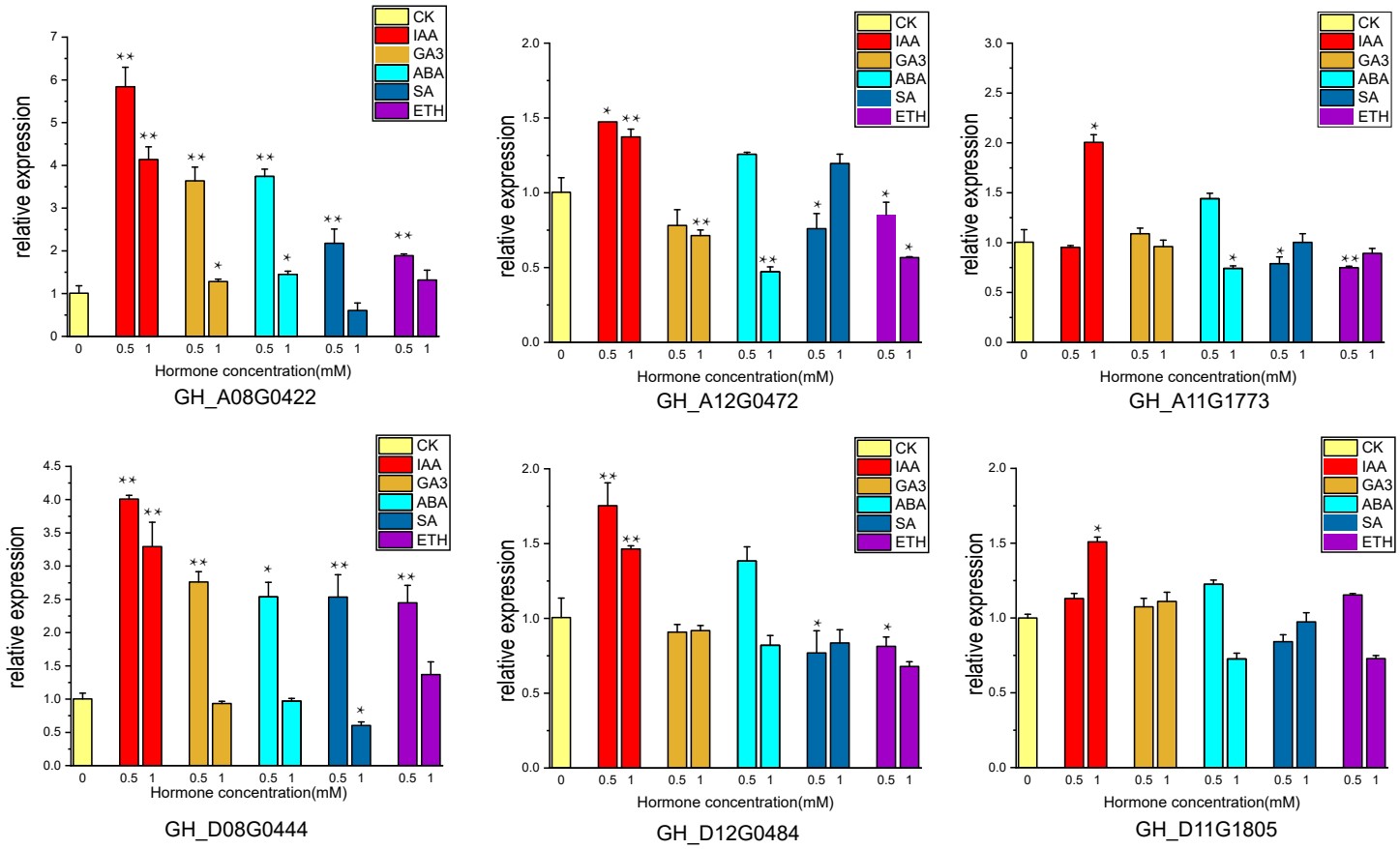

**Figure 9** The relative expression levels of the 6 genes in the ovule at 15 DPA after hormone treatment were verified by qRT-PCR. CK is controlled without additional hormones, and the ovules only grow on BT medium. *, $P < 0.05$; **, $P < 0.01$.

RNA extraction, and the expression of six highly expressed genes under the influence of five hormones was measured, and it was found that the expression of *UGP* genes increased correspondingly under the influence of IAA and GA3, and the expression of *GH_A08G0422* and *GH_D08G0444* increased 6-fold and 4.5-fold, respectively, and the expression of these two genes increased under the influence of low concentrations of ABA, ETH and SA also increased the expression of these two genes under the influence of low concentrations of ABA, ETH and SA. It showed that expression of *UGP* genes was influenced by hormones and was more strongly stimulated for growth hormone and gibberellin (*Bai et al., 2014*; *Xiao, Zhao & Zhang, 2019*). It is tentatively hypothesized that *UGP* genes have an indispensable role in cotton fiber development. Its expression is closely related to the activity of pectin pathway in the fiber.

## CONCLUSION

In this study, a total of 66 *UGP* genes were identified based on the genomic information of *G. raimondii*, *G. arboreum*, *G. hirsutum* and *G. barbadense*. Covariance analysis postulated that the amplification of *UGP* genes was due to repetitive tandem generation of new family

genes. Ka/Ks ratio analysis postulated that *UGP* genes are under strong purifying selection pressure in cotton. Six highly expressed genes namely *GH_A08G0422*, *GH_D08G0444*, *GH_A11G1773*, *GH_D11G1805*, *GH_A12G0472* and *GH_D12G0484*, all possessing relatively long UDPGP structural domains, were obtained by qRT-PCR and transcriptome data screening of cotton fiber. The addition of 0.5 mM IAA and GA3 to ovule culture medium promoted the growth of ovule fiber clusters, which showed an increase of about 10% in area and the expression of six *UGP* genes increased from 1.5-fold to 6-fold. These results suggest that *UGP* genes may play an important role in the growth and development of cotton fibers, and that the speed of cotton fiber development and level of *UGP* gene expression are closely related, and that their mechanisms of action in cotton fiber development and fiber quality formation and their effects on pectin synthesis in cotton fiber development deserve in-depth study.

### Funding
This work was funded by the Natural Science Foundation of China (NO.3210161266). The funders had no role in study design, data collection and analysis, decision to publish, or preparation of the manuscript.

### Grant Disclosures
The following grant information was disclosed by the authors:
The Natural Science Foundation of China: NO.3210161266.

### Competing Interests
Tehseen Azhar Muhammad is an Academic Editor for PeerJ.

### Author Contributions

- Zhongyang Xu conceived and designed the experiments, performed the experiments, analyzed the data, prepared figures and/or tables, authored or reviewed drafts of the article, and approved the final draft.
- Jiasen He analyzed the data, authored or reviewed drafts of the article, and approved the final draft.
- Muhammad Tehseen Azhar analyzed the data, authored or reviewed drafts of the article, and approved the final draft.
- Zhen Zhang analyzed the data, authored or reviewed drafts of the article, and approved the final draft.
- Senmiao Fan analyzed the data, authored or reviewed drafts of the article, and approved the final draft.
- Xiao Jiang analyzed the data, authored or reviewed drafts of the article, and approved the final draft.
- Tingting Jia performed the experiments, authored or reviewed drafts of the article, and approved the final draft.

- Haihong Shang conceived and designed the experiments, authored or reviewed drafts of the article, and approved the final draft.
- Youlu Yuan conceived and designed the experiments, authored or reviewed drafts of the article, and approved the final draft.

## Data Availability

The raw measurements are available in the Supplementary Files.

## Supplemental Information

Supplemental information for this article can be found online at http://dx.doi.org/10.7717/peerj.13460#supplemental-information.

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
