# Peer review of "UDP-glucose pyrophosphorylase: genome-wide identification, expression and functional analyses in Gossypium hirsutum"

_PeerJ, doi:10.7717/peerj.13460_

## Round 0.1 · original submission · Major Revisions

Dear authors,

Your article has been reviewed by 3 peer reviewers. The reviewers have provided evaluations and made recommendations for revisions to your manuscript. All of the reviewers have raised some concerns about your interpretations, which need to be carefully considered.

I invite you to respond to the reviewers' detailed comments and revise your manuscript. All three reviewers' comments need to be addressed before the manuscript can be accepted.

Thank you for submitting your manuscript to PeerJ and I look forward to receiving your revision.

Best,

Xiaotian Tang, PhD
Academic Editor, PeerJ
xiaotian.tang@yale.edu

Reviewer 1 ·

Basic reporting

In this manuscript, the authors identified 66 UGP genes in four cultivated cotton species. For the widely-cultivated Gossypium hirsutum, a total of 19 genes were found. The transcriptome data and qRT-PCR results showed that six genes from G. hirsutum were highly expressed in 15-DPA fiber. Furthermore, the authors studied their potential function in fiber development with different sorts and different gradients of phytohormones by ovule culture assays. They found that low IAA and GA3 concentrations promoted the development of ovules and fiber clusters, and the expressions of all six UGP genes were increased to different degrees in the 15 DPA fibers from ovule culture. These results provide insights into some of the UGP genes that may influence cotton fiber development. However, some changes are needed to improve the quality of this manuscript.
1. Please revise the abstract because of large proportion about the UGP family from four cotton species. Or the title of this manuscript should be revised to “UDP-glucose pyrophosphorylase: genome-wide identification, expression and function analyses in cotton” or other better title the authors think.
2. L139, L153: the sentences need to be rewritten.
3. In the Methods, part of the description in lines 177-187 is too similar to another paper (UDP-Glucose Dehydrogenases: Identification, Expression, and Function Analyses in Upland Cotton (Gossypium hirsutum), Tingting jia, 2020).
4. In the Result, 3.2 developmental is improper. The title can be revised to “Evolutional analysis of UGP gene family”. 3.3 subpopulation should be subgroup.
5. L 228: their is redundant.
6. L233-L236: Please carefully check the sentences.
7. Fig.8B and Fig.9 are lack of the statistical significance analyses.
8. In the Conclusion, the author restates the main results. Please rewrite the conclusion.
9. land cotton should be upland cotton. Conventionally, L145 “BP” should be bp. L288 “p<0.05” should be P<0.05. G. arboretum should be revised to G. arboreum.
10. There were numerous typos and grammatical problems in the text that would require thorough English editing prior to the formal publication.

Experimental design

no comment

Validity of the findings

no comment

Additional comments

no comment

Reviewer 2 ·

Basic reporting

In this study, a total of 66 UDP-glucose pyrophosphorylase (UGP) (EC 2.7.7.9) genes were identified from the genomes of four cotton species. The analysis of its evolutionary relationship, gene structure, and expression provides a basis for further studies on the function of UGP genes in cotton development. The manuscript can be considered for publishing in Peer J after moderate revision.

Some sentences are very confusing and lots of grammar mistakes need to be fixed.
Line 97, the round brackets should not be italic.
Line 122, “G. arboretum” should be “G. arboreum”.
Line 139-141 should be adjusted.
Line 153 is not a complete sentence.
Line 154-160. The sentence is too long and confused.
……
The English language should be improved to ensure that an international audience can clearly understand your text. I suggest authors have a colleague who is proficient in English and familiar with the subject matter review your manuscript, or contact a professional editing service.

The format of references should be adjusted according to the Journal’s requirement, e.g., some Journals’ names are in full name, whereas some others are in abbreviation; Lattin name of species should be italic.

Lattin name in the legend of Figure 1 should be italic. The legend of Figure 2 should be more explicit.

In the tables, “table 1” should be “Table 1”, et al. Gene names in Tables should be italic.

Experimental design

The novelty and experimental design of the manuscript is fine.

Validity of the findings

The authors validate the evolution and expression of the UGP gene family in cotton.

Additional comments

no comment.

Reviewer 3 ·

Basic reporting

The language should be improved since some statements were quite confused, for example, the UGP genes in some species such as Arabidopsis has been reported, but what you describe in your article is' A total of 81 UGP genes were identified in seven species' (line 192) or ‘we identified 81 UGP genes in seven different species’ (line 297).

Experimental design

Statistical analysis need to be shown in all figures and the methods for data analysis should be supplemented in ‘Materials and Method’.

Validity of the findings

no comment

Additional comments

I found that the phylogenetic tree was quite different from the previous report (Wang et al, Plant Cell Reports, 2011). In previous study, the UGP genes were divided into five subfamilies in cotton, which was inconsistent with this manuscript, the authors should explain that, or at least, discuss it.

The figure legends are not detailed enough. The ordinate title was missing in figure 9.

---

## Round 0.2 · accepted · Accept

Dear Authors,

Thank you for adequately addressing all the concerns raised during the initial review. I am pleased to inform you that your article, "UDP-glucose pyrophosphorylase: genome-wide identification, expression and functional analyseis in Gossypium hirsutum", has now been accepted for publication in PeerJ. Congratulations!

Thank you for your submission and we hope you will continue to support PeerJ.

Best,

Xiaotian Tang
Academic Editor, PeerJ
xiaotian.tang@yale.edu